# Toxicokinetics of Zearalenone following Oral Administration in Female Dezhou Donkeys

**DOI:** 10.3390/toxins16010051

**Published:** 2024-01-17

**Authors:** Honglei Qu, Yunduo Zheng, Ruifen Kang, Yulong Feng, Pengshuai Li, Yantao Wang, Jie Cheng, Cheng Ji, Wenqiong Chai, Qiugang Ma

**Affiliations:** 1State Key Laboratory of Animal Nutrition and Feeding, College of Animal Science and Technology, China Agricultural University, Beijing 100193, China; leihong_qu@163.com (H.Q.); zhengyunduo@cau.edu.cn (Y.Z.); ruifenkang@cau.edu.cn (R.K.); lpsmyj@cau.edu.cn (P.L.); jicheng@cau.edu.cn (C.J.); 2National Engineering Research Center for Gelatin-Based Traditional Chinese Medicine, Dong-E-E-Jiao Co., Ltd., Liaocheng 252201, China; fengyulong19871024@163.com (Y.F.); wangyt@dongeejiao.com (Y.W.); chengjie@dongeejiao.com (J.C.); 3Liaocheng Research Institute of Donkey High-Efficiency Breeding and Ecological Feeding, Liaocheng University, Liaocheng 252000, China

**Keywords:** toxicokinetic, donkey, mycotoxins, zearalenone

## Abstract

Zearalenone (ZEN) is a mycotoxin produced by various Fusarium strains, that is present in food and feed raw materials worldwide, causing toxicity effects in animals and humans. This research aimed to explore the toxicokinetics of ZEN on female Dezhou donkeys following a single oral exposure dosage of 2 mg/kg BW (body weight). The sample collection of donkeys plasma was carried out at 0, 5, 10, 15, 20, 30, 45, 60, 90 min, 2 h, 2.5 h, 3 h, 3.5 h, 4 h, 4.5 h, 6 h, 9 h, 12 h, 24 h, 48 h, 72 h, 96 h and 120 h via intravenous catheter, and fecal and urinary samples were severally collected at 0 h and every 6 h until 120 h. The concentrations of ZEN, α-zearalenol (α-ZOL), β-zearalenol (β-ZOL), α-zearalanol (α-ZAL), β-zearalanol (β-ZAL), zearalanone (ZAN) in plasma, urine, and feces were detected by UPLC-MS/MS. Only ZEN was detected in plasma, and the maximum was 15.34 ± 5.12 µg/L occurred at 0.48 h after gavage. The total plasma clearance (Cl) of ZEN was 95.20 ± 8.01 L·kg·BW^−1^·h^−1^. In addition, the volume of distribution (Vd) was up to 216.17 ± 58.71 L/kg. The percentage of total ZEN (ZEN plus the main metabolites) excretion in feces and urine was 2.49% and 2.10%, respectively. In summary, ZEN was fast absorbed and relatively slowly excreted in female donkeys during 120 h after a single gavage, indicating a trend of wider tissue distribution and longer tissue persistence.

## 1. Introduction

Zearalenone (ZEN) is an estrogen-like mycotoxin produced by a variety of genus Fusarium [1]. ZEN is widely present in critical grains such as maize, wheat, oats, barley, and their byproducts [2]. In a 10-year global investigation of mycotoxin contamination, the positive ratio of ZEN was up to 45% in a total of 61,413 samples [3]. ZEN was also one of the highest occurring mycotoxins in China, according to 9392 samples tested from 2017 to 2021, with a positive detection rate of 33.42–79.09% and average positive concentration of 74.20–300.57 µg/kg respectively in different years [4]. Although ZEN has a large lactone ring, ZEN is a heat-stable toxin with a relatively high melting point (164 to 165 °C) [5,6]. In natural conditions, the degradation of ZEN can only be observed at very high temperatures or in an alkaline environment, which makes ZEN remain stable during storage, transportation, milling, and processing [7,8]. Therefore, ZEN can be transferred through the food chain to affect the health of animals and humans [9,10]. ZEN can cause reproductive disorders in different animals due to the similar chemical structure to estrogen. Numerous types of research have shown the adverse effect of reproduction induced by ZEN, including the swollen vulva, vaginal prolapse, pseudopregnancy, infertility, and abortion [11,12,13,14]. Moreover, studies have shown that ZEN also could inhibit biomolecule synthesis and cell proliferation in different cell lines [15]. In addition, the treatment of first-parity gestation sows with 246 µg/kg ZEN significantly increased cell apoptosis in organs and oxidative stress [16]. ZEN impaired immune defense after pigs were given a 1678.42 µg/kg ZEN-containing diet for two weeks, inflammatory lesions of immune organs (spleen and thymus), adverse oxidative stress (increased mRNA expression of GSH-Px, Cu/Zn-SOD, and Mn-SOD), mRNA expression of proinflammatory cytokine in liver, thymus, and uterus increased [17]. The treatment of rats with 1.0 and 5.0 mg/kg ZEN for four weeks injured gut morphology, increased intestinal permeability and decreased the expression of mucin and tight junction proteins [18]. These research suggest that ZEN is widespread in foods or feed commodities and cause adverse effects in animals, such as reproductive toxicity, immunotoxicity, enterotoxicity, and cytotoxicity.

Once ingested by the animal, ZEN is rapidly and extensively absorbed via the gastrointestinal tract, and then distributed to target organs [19]. Pigs are the most sensitive animals to ZEN toxicity, followed by rats, poultry, and ruminants, which may be attributable to distinctive mechanisms of absorption, metabolism, distribution, and excretion of ZEN in various animals. Furthermore, the pharmacokinetics of ZEN have been studied on different types of animals by oral or intravenous injection, involving pigs [20,21], goats [22], rats [23,24,25], broiler chickens, turkey poults and laying hens [26,27,28]. These previous toxicokinetic studies showed that the time to maximum plasma ZEN concentration (Tmax) following oral in pigs is approximately 0.25–2 h, which was longer than other animals [19,29]. Meanwhile, The elimination half-life (T1/2) of ZEN in pigs was about 25–86.8 h, which was much longer than in rats [23,24,25], and chickens [26,27,28]. Therefore, ZEN toxicokinetic parameters are related to the sensitivity of specific animals to the specific toxin, which is essential to evaluate its adverse effects.

Donkeys have primarily been utilized as agricultural labor for thousands of years. In recent years, with the modernization of agriculture, donkeys have emerged as multi-purpose farm creatures in some countries, such as companion animals, and even dairy or meat-producing animals [30]. Our previous toxicokinetics research of ochratoxin and deoxynivalenol in donkeys showed the absorption, elimination, and excretion parameters of both toxins in donkeys were different from other animals [31,32]. However, there is no research to be found about the toxicokinetics of ZEN in donkeys. Therefore, the current study was designed to investigate the toxicokinetics of ZEN following a single oral dose of ZEN in female DeZhou donkeys.

## 2. Results

### 2.1. Method Validation

Figure 1 shows the total ion chromatograms of ZEN and its five metabolites mixed standards in four matrices (methanol, plasma, urine, and feces). The matrix influence was acceptable for 6 target substances according to the relatively separate and defined retention times. As reflected in Figure 2, the calibration plot indicated linear tendencies in the range of 1.25–25 µg/L in plasma, and the typical coefficient of determinations (R^2^) above 0.9921 was found in both ZEN and its metabolites. For ZEN and metabolites of feces and urine, the linear tendencies were obtained in the range of 1.25–500 (µg/kg or µg/L) with a good coefficient of determinations (R^2^) above 0.9925, and 0.9985, respectively. In addition, as shown in Table 1, the limits of detection (LOD) for ZEN and its metabolites in plasma were range of 0.5–1.5 µg/L, and the limits of quantification (LOQ) were range of 1.5–4.5 µg/L. For ZEN and its metabolites in feces, the range of LOD and LOQ was 0.3–1.0 µg/kg and 1.0–3.0 µg/kg, respectively. The LOD and LOQ of ZEN and its metabolites in urine were in a range of 0.5–2 µg/L and 1.5–6.0 µg/L, singly. Then, two spike levels were chosen to assess the recovery rate in plasma (2.5, 25 µg/L), feces (5, 100 µg/kg), and urine (5, 100 µg/L), which refer to low and high concentrations. The mean recovery rates of ZEN and its metabolites ranged from 74.04–92.95%, 70.99–88.12%, and 73.35–93.15% in plasma, feces, and urine, respectively (Table 2).

### 2.2. Toxicokinetic Parameters of ZEN in Female Donkey Plasma

The toxin was absorbed into the blood circulation of donkeys following the oral exposure (2000 µg/kg BW). ZEN was detected in the plasma samples. As shown in Table 3, the plasma levels of ZEN reached a peak concentration (Cmax) at 0.48 ± 0.10 h (Tmax) after oral gavage. The elimination of half-life was 1.63 ± 0.46 h. The volume of distribution was 216.17 ± 58.71 L·kg·BW^−1^.

### 2.3. Plasma Concentration of ZEN

As shown in Figure 3, the ZEN in plasma was first detected at 5 min after a single oral administration, and the concentration of ZEN increased rapidly until the peak level was reached at 0.48 ± 0.10 h, then gradually decreased. ZEN could not be detected at 4.5 h after oral gavage. The metabolites of ZEN, including α-ZOL, β-ZOL, α-ZAL, β-ZAL, and ZAN were not detected in plasma.

### 2.4. Recovery of ZEN and Its Metabolites Excreted in Feces and Urine

As shown in Figure 4A–C, ZEN, α-ZOL, and β-ZOL were detected in the feces respectively at 18 h after oral administration with similar elimination characteristics. The average excretions increased promptly from 18 h to 60 h, afterward decreasing gradually until 102 h when low levels of ZEN, α-ZOL, and β-ZOL could be detected in feces. Moreover, after 18 h of ZEN oral, α-ZOL and β-ZOL could be the two main ways of toxin elimination that were higher than ZEN excretions (Figure 4D).

As shown in Figure 5, ZEN-containing urine was first excreted 6 h after oral administration. The metabolites of ZEN, namely α-ZOL and β-ZOL were detected in the urine. The amount of ZEN, α-ZOL, and β-ZOL average excretion have similar curves, the elimination rapidly rose between 6 h and 30 h, then decreased gradually until low levels of ZEN, α-ZOL, and β-ZOL could be detected at 102 h. In addition, β-ZOL was the major excreted pathway in urine after 24 h of ZEN oral administration in donkeys.

Feces and urine are the two primary pathways to eliminate toxins in animals and humans. As presented in Table 4, donkeys were administered a total of 309.75 ± 9.53 mg of ZEN, with 2.49 ± 0.43% excreted in the feces and 2.10 ± 0.46% excreted in the urine. The rate of absorption was 97.51 ± 0.43%. These values indicate that donkeys have a high absorption rate of ZEN and that feces are the major way of ZEN elimination.

## 3. Discussion

The toxicokinetics of ZEN primarily involves the entry rate of ZEN into the body, absorption, distribution, metabolism, and elimination, and varies in different animals. In the present study, donkeys were used as research objects due to their important function in providing farm labor, transportation of goods, companions, meat, milk, and traditional medicines for humans. Nevertheless, there are no available studies on the toxicokinetics of ZEN in donkeys by oral or intravenous injection. Therefore, the current study aimed to clarify the toxicokinetic of ZEN following single oral gavage in donkeys.

After oral administration, ZEN is absorbed swiftly and extensively by the gastrointestinal tract [9]. Absorption means the amount and time required for the toxin to reach the plasma from the route of administration, and Tmax is the major parameter for measuring absorption, which represents the rate of absorption [33]. The Tmax was reported to be 0.25 h for rats fed 3 mg ZEN/kg BW and 8 mg ZEN/kg BW respectively [23,24]. After oral giving 3 mg ZEN/kg BW to Ross 308 broilers, the Tmax was 0.35 h [28]. In addition, after a one-time oral gavage of 1.2 mg ZEN/kg BW broiler chicken, the peak time of plasma concentration was detected at 0.25 h [26]. The study results laying Hens and Turkey showed the Tmax of following a single oral dosage of 3 mg/kg ZEN was 0.32 h and 0.97 h separately [28]., The Tmax was reported to be 0.5–2 h for pigs after a single oral dose of 1 mg ZEN/kg BW [20,29], which indicates the Tmax of pigs is much longer than that in rats, broiler, and laying hens. Moreover, the terminal elimination half-life (T_1/2_Elim) is the typical parameter of excretion. The T_1/2_Elim value of ZEN after oral was 5.6–16.8 h in rats [23,24,25], 0.34 h in the broiler, 0.36 h in laying hens and 0.35 h in turkey poults [28], the elimination rate of which was more rapid than pigs [34]. The high absorption with slow elimination of ZEN in pigs may be one of the causes for the greater susceptibility. In the present toxicokinetics study of donkeys, Tmax after an oral dose of 2 mg/kg BW was 0.48 ± 0.10 h, that value was lower than in pigs but higher than in rats, broilers, laying hens, and turkeys. Meanwhile, the half-life elimination of donkeys was 1.63 ± 0.46 h, which was longer than chickens, whereas shorter than pigs and rats. The results indicate that donkeys have a moderate rate of ZEN elimination as well as absorption and tend to accumulate in vivo. We can conjecture that donkeys may be less susceptible to ZEN than pigs and rats, and more sensitive than broilers, laying hens, and turkeys, but that needs further research to verify.

After rapid absorption, the toxin will be distributed to the tissues and organs with blood circulation. Distribution refers to the toxin positioning in the body, which was measured by the volume of distribution (Vd). A large Vd value often means wide systemic exposure and prolonged tissue persistence. Previous studies have shown that the Vd value of pigs was 7.27–99 L/kg [29]. While the Vd values of rats, broilers, laying hens, turkey poults, and goats were respectively 2.0–4.7 L/kg, 3.2–22.6 L/kg, 6.18–6.24 L/kg, 10.65 L/kg and 7.32 L/kg [24,25,26,27], that were obviously lower than pigs. In the present study, the Vd value was up to 216.17 ± 58.71 L/kg after the single oral dose of 2 mg/kg BW ZEN in donkeys, which may imply a wider tissue distribution. In addition, feces and urine are the two main routes of toxin elimination in animals after the metabolism in the tract and liver. Rats excrete ZEN via both feces and urine after oral administration of 1 and 10 mg/kg BW, the percentages were severally about 55% and 15–20% [35]. Analogous to rats, after approximately oral administration of 6 µg/kg BW to broilers, accumulated excretion of both ZEN and α-ZOL amounted to approximately 58% of total intake after 48 h [36]. Similar to broilers, about 94% of ^14^C-labelled ZEN was eliminated through the excreta within 72 h of dosing to laying hens [37]. Meanwhile, following oral gavage ZEN (10 µg/kg BW) to piglets, the biological recovery in urine was 26 ± 10%, feces 14 ± 3% during 48 h, and the total excreta was 40 ± 8% [38]. However, in the present research, only 2.10% of ZEN was excreted by urine, and only 2.49% was excreted by feces for 5 days. The amount of ZEN excreted by donkeys was less than in other animals which may be related to the absorption and distribution patterns, the size of the animals, the route of exposure, and toxin dosage.

## 4. Conclusions

The present research indicated that after oral gavage of 2 mg ZEN/kg BW to female donkeys, the ZEN in plasma reached a maximum of 0.48 h, and the terminal elimination half-life was 1.63 h. Meanwhile, the volume of distribution was 216.17 L/kg, indicating that ZEN has a trend of wide tissue distribution and prolonged tissue persistence. In addition, the amount of excretion in feces and urine was approximately 4.59% of the ZEN intake during 120 h. Nevertheless, the biotransformation of ZEN in the gastrointestinal tract and liver of donkeys remains unclear, and the specific effects and dosage effects in donkeys should be further researched in the future.

## 5. Materials and Methods

### 5.1. Chemicals, Products and Reagents

The standards of ZEN, α-ZOL, β-ZOL, α-ZAL, β-ZAL, and ZAN used for animal experiments and sample analysis were obtained from Pribolab Biological Engineering Co., Ltd. (Qingdao, China). Water, acetonitrile (ACN), and methanol (MeOH) used for the sample analysis were all HPLC-MS grade (Merck, Darmstadt, Germany). Dimethyl sulfoxide (DMSO) were purchased from Solarbio (Beijing, China) and physiological saline used for the animal oral administration were both cell-culture grade.

### 5.2. Animal and Treatment

Four healthy 9-month-old Dezhou female donkeys (154.88 ± 4.76) were chosen and housed individually in metabolism cages, which were designed to ensure the collection all of fecal and urine samples, as well as a proper space for the movement of donkeys. The blank blood, feces, and urine samples were obtained 4 h before the beginning of the experiment. The same feeding and watering strategies were performed during the 5-day adaptation period and 5-day experiment. The oral solution was prepared by dissolving ZEN standard in DMSO (stock solution, 10 mg/mL), then diluted to work solution (2 mg/mL) with physiological saline. ZEN working solution was administered by gavage tube in a single dosage of 2 mg ZEN/kg BW. The study was approved by the Laboratory Animal Welfare and Animal Experimental Ethical Committee of China Agricultural University (No. AW80803202-1-8).

### 5.3. Samples Collection

The blood samples were obtained from the jugular vein in donkeys (post-hepatic) before administration (0 min) and 5, 10, 15, 20, 30, 45, 60, 90 min, 2 h, 2.5 h, 3 h, 3.5 h, 4 h, 4.5 h, 6 h, 9 h, 12 h, 24 h, 48 h, 72 h, 96 h and 120 h after ZEN oral gavage. The samples in heparin anticoagulation tubes were transferred to the laboratory and then centrifuged at 3000 rpm, for 15 min to obtain plasma. Moreover, fecal, and urinary samples were individually gathered before administration (0 h) and 6, 12, 18, 24, 30, 36, 42, 48, 54, 60, 66, 72, 78, 84, 90, 96, 102, 108, 114, 120 h after the oral of ZEN. Meanwhile, the feces weight and urine volume were recorded at each collection time. All samples were stored at −20 °C for further analysis.

### 5.4. Sample Treatment

Plasma and urine were thawed completely, and feces were lyophilized and homogenized, 1 mL plasma, 2 mL urine, and 2 g feces samples were transferred to 50 mL tubes, individually. Then, acetonitrile/water (80/20, *v*/*v*) was added into the tubes with 20 mL for feces and urine, and 15 mL for plasma. The tubes were vortexed for 2 min, sonicated for 1 h, and centrifuged for 5 min at 8000 rpm. 2 mL feces or urine treated supernatant, 3 mL plasma treated supernatant was pipetted into 15 mL centrifuge tubes and evaporated to dryness using nitrogen at 40 °C. Then, the samples were reconstituted with acetonitrile/water (10/90, *v*/*v*) and up to a final volume of 1 mL. All samples were filtered using a 0.22 µm filter after 1 min vortex.

### 5.5. Standard Solutions

Standard stock solutions were prepared firstly, 5 mg of ZEN, α-ZOL, β-ZOL, α-ZAL, β-ZAL, and ZAN were dissolved in 1 mL of methanol, respectively. Then, each stock solution was diluted by acetonitrile/water (50/50, *v*/*v*) to individual concentrations of working solutions (0.0125, 0.025, 0.05, 0.125, 0.25, 0.5, 1, 2.5, 5 µg/mL). 10 µL working solution was added to 90 µL control plasma and urine samples, individually, which were used to take five levels of plasma spiked samples (1.25–25 µg/L) and nine levels of urine spiked samples (1.25–500 µg/L). Meanwhile, nine spiked feces samples range of 1.25–500 µg/L were made by adding 100 µL working solutions to 1 g control fecal samples.

### 5.6. Method Validation

The method was validated on linearity, sensitivity, and recovery rate in plasma, urine, and feces, separately. Calibration curves were obtained by spiking control substrates (plasma, feces, and urine) with different levels of ZEN and metabolites. LOD and LOQ were applied to evaluate the sensitivity, which was signal-to-noise ratio (S/N) ≥ 3 for LOD, and S/N ≥ 10 for LOQ. Recoveries were obtained by comparing the peak areas of spiked samples in the matrix (plasma, feces, urine) with the peak areas of the matching standard working solutions.

### 5.7. UPLC-MS/MS Analysis

The ACQUITY UPLC BEH C18 (100 mm × 2.1 mm, 1.7 μm) column (Waters, Framingham, MA, USA) was prepared for the separation. The injection volume was 2.0 µL, mobile phase A consisted of 0.1% formic acid in water, while mobile phase B was ACN at a flow rate of 0.3 mL/min. The gradient elution procedure was performed as follows: 0–2 min 15% B; 2–4 min 40% B; 4–6.5 min 40% B; 6.5–9 min 60% B; 9–12.5 min 60% B; 12.5–13.0 min 15% B; followed by a re-equilibration time of 2 min. The total run time was 15 min. ZEN and its metabolites were quantified via ultra-performance liquid chromatography-triple quadrupole mass spectrometer (AB Sciex 5500, Framingham, MA, USA). The electrospray interface (ESI) conditions were as follows: ion spray voltage (IS) at 5.5 kV; curtain gas (CUR) at 20 psi; nebulizer gas (GS1) at 55 psi and ion source temperature at 450 °C.

### 5.8. Statistical Analysis

The standardized concentrations of ZEN and metabolites were used to perform statistical analysis. Plasma toxicokinetic parameters of ZEN were calculated using the non-compartmental modeling in WinNonlin 5.2.1 software (Certara, Inc., Princeton, NJ, USA). Then the average ZEN concentrations in plasma at different times were used to chart the plasma concentration-time profiles. Mean fecal and urinary excretion of ZEN and metabolites at different times were used to chart the corresponding excretion–time profiles separately. The figures were charted using GraphPad Prism version 9.4.1 (GraphPad Software, Inc., San Diego, CA, USA). Data are shown as mean ± SEM.

## Figures and Tables

**Figure 1 toxins-16-00051-f001:**
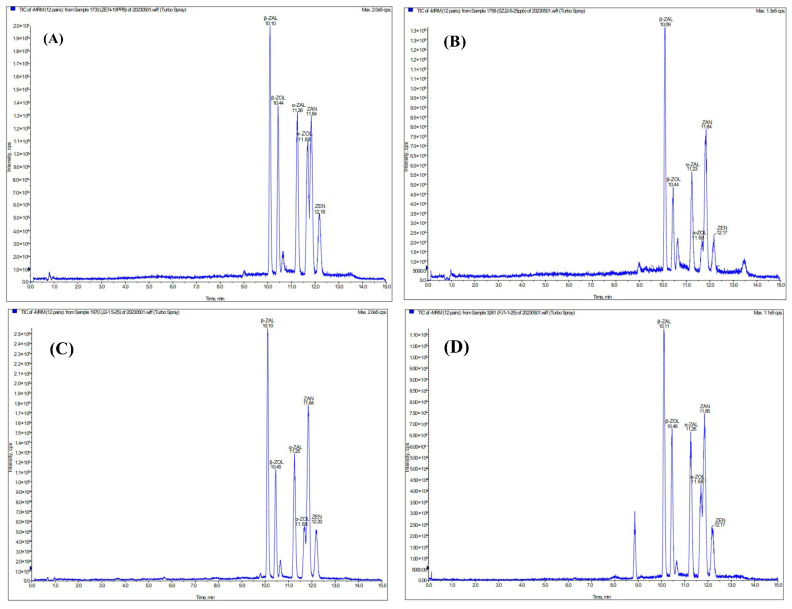
Total ion chromatograms of ZEN and five metabolite mixed standards in different matrices. (**A**): Methanol solution as a matrix (10 µg/L for each target substance), (**B**): Plasm as a matrix (25 µg/L for each target substance), (**C**): Urine as a matrix (25 µg/L for each target substance), (**D**): Feces as a matrix (25 µg/L for each target substance).

**Figure 2 toxins-16-00051-f002:**
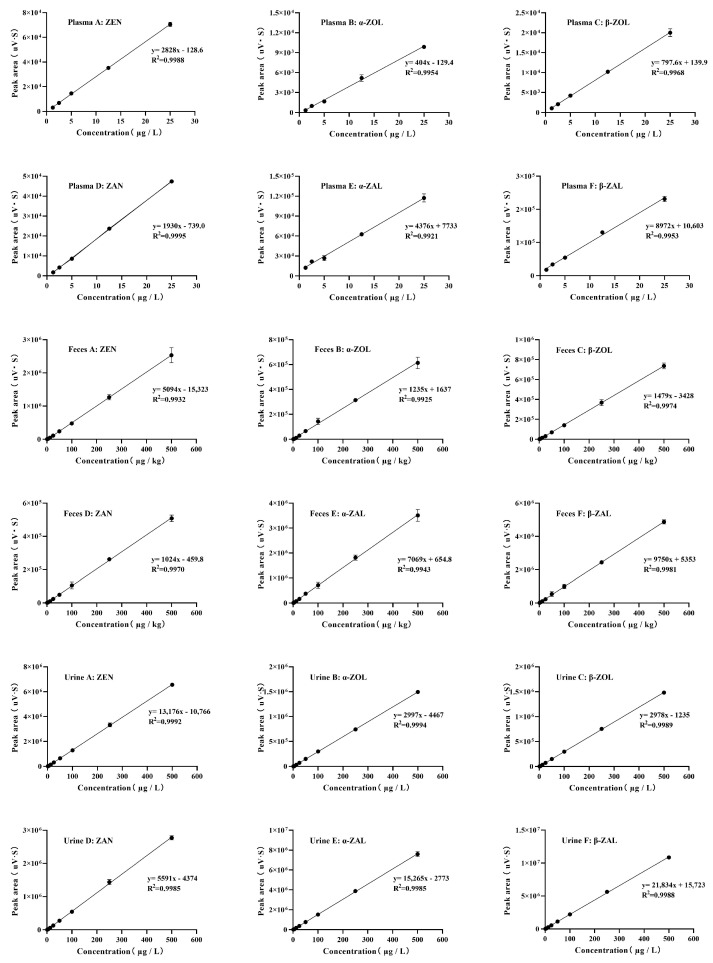
Calibration plots for adding different levels of standard samples, *n* = 3. Plasma (A–F): 1.25, 2.5, 5, 12.5, 25 µg/L; Feces (A–F): 1.25, 2.5, 5, 12.5, 25, 50, 100, 250, 500 µg/kg; Urine (A–F): 1.25, 2.5, 5, 12.5, 25, 50, 100, 250, 500 µg/L.

**Figure 3 toxins-16-00051-f003:**
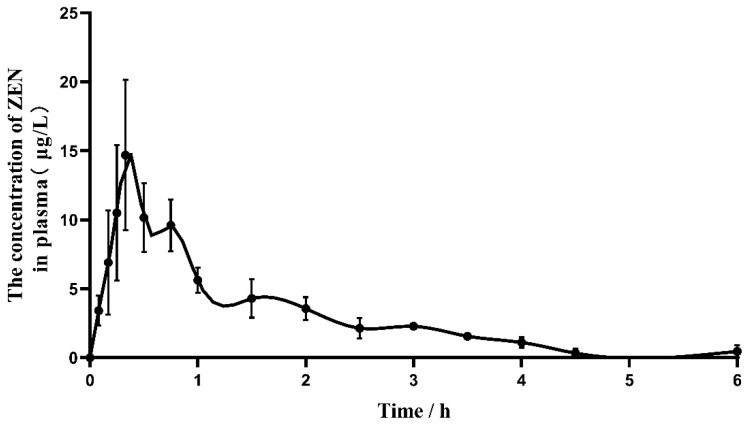
Plasma concentration-time profiles for zearalenone (ZEN) in donkeys after oral gavage of 2000 µg/kg· BW, *n* = 4.

**Figure 4 toxins-16-00051-f004:**
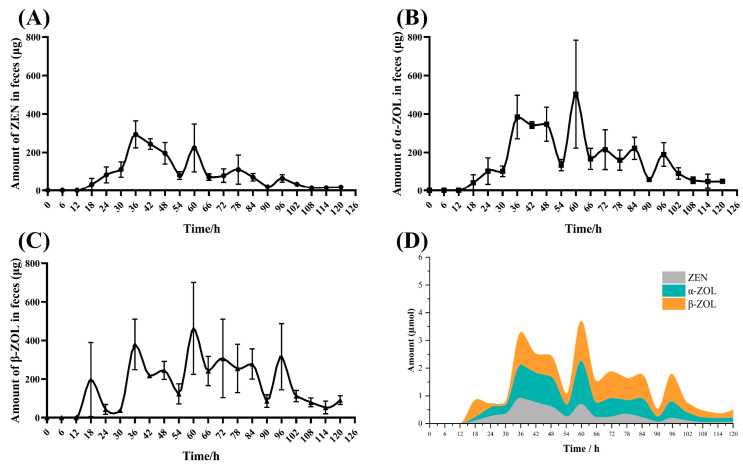
The excretion of zearalenone and metabolites in the feces of donkeys following single oral dose of 2000 µg/kg· BW, *n* = 4. (**A**): The average of ZEN elimination—time curve in feces, (**B**): The average of α-ZOL elimination—time curve in feces, (**C**): The average of β-ZOL elimination—time curve in feces, (**D**): The stacked area curve of ZEN and metabolites elimination (µ mol) in feces.

**Figure 5 toxins-16-00051-f005:**
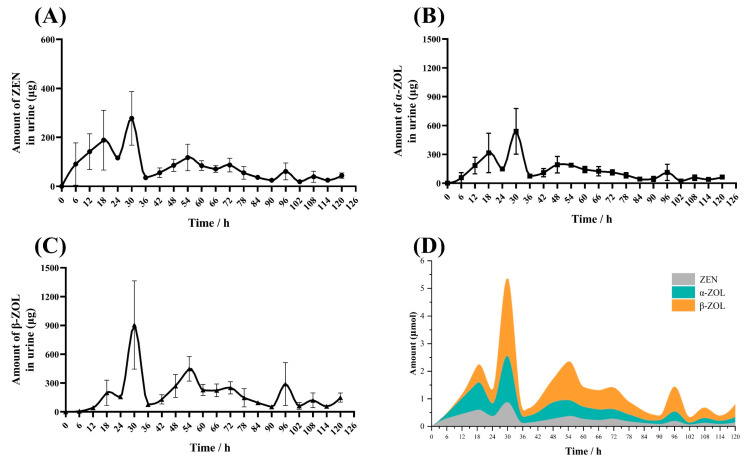
The average excretion of zearalenone and metabolites in the urine of donkeys following single oral dose of 2000 µg/kg·BW, *n* = 4. (**A**): The average of ZEN elimination—time curve in urine, (**B**): The average of α-ZOL elimination—time curve in urine, (**C**): The average of β-ZOL elimination—time curve in urine, (**D**): The stacked area curve of ZEN and metabolites elimination (µ mol) in urine.

**Table 1 toxins-16-00051-t001:** Calibration plots of ZEN and metabolites in plasma, feces, and urine.

Matrix	Composition	Slope	R^2^	Range (µg/L)/(µg/kg)	Sensitivity (µg/L)/(µg/kg)	
					LOD	LOQ
Plasma	ZEN	2828	0.9988	1.25–25	1.5	4.5
	α-ZOL	404.0	0.9954	1.25–25	1.0	3.0
	β-ZOL	797.6	0.9968	1.25–25	1.5	4.5
	α-ZAL	4376	0.9921	1.25–25	0.5	1.5
	β-ZAL	8972	0.9953	1.25–25	0.5	1.5
	ZAN	1930	0.9995	1.25–25	1.5	4.5
Feces	ZEN	5094	0.9932	1.25–500	1.0	3.0
	α-ZOL	1235	0.9925	1.25–500	0.3	1.0
	β-ZOL	1479	0.9974	1.25–500	0.5	1.5
	α-ZAL	7069	0.9943	1.25–500	1.0	3.0
	β-ZAL	9750	0.9981	1.25–500	1.0	3.0
	ZAN	1024	0.9970	1.25–500	0.3	1.0
Urine	ZEN	13,176	0.9992	1.25–500	1.5	4.5
	α-ZOL	2997	0.9994	1.25–500	1.0	3.0
	β-ZOL	2978	0.9989	1.25–500	2.0	6.0
	α-ZAL	15,265	0.9985	1.25–500	0.5	1.5
	β-ZAL	21,834	0.9988	1.25–500	0.5	1.5
	ZAN	5591	0.9985	1.25–500	2.0	6.0

*n* = 3 per concentration, µg/kg refers to the values of feces; µg/L refers to the values of plasma and urine.

**Table 2 toxins-16-00051-t002:** Recovery of ZEN and metabolites for plasma, feces, and urine.

Item	Plasma Spike Level (µg/L)	Plasma Recovery (%)	Feces and Urine Spike Level (µg/L)/(µg/kg)	Feces Recovery (%)	Urine Recovery (%)
ZEN	2.5	74.04	5.0	88.12	78.78
	25	79.20	100	82.46	75.39
α-ZOL	2.5	88.06	5.0	83.98	93.15
	25	78.65	100	71.46	75.86
β-ZOL	2.5	92.95	5.0	80.49	82.27
	25	84.11	100	74.72	73.35
α-ZAL	2.5	95.44	5.0	75.13	85.41
	25	76.22	100	83.91	80.25
β-ZAL	2.5	90.97	5.0	70.99	74.98
	25	78.58	100	79.31	86.64
ZAN	2.5	91.14	5.0	80.97	86.85
	25	88.94	100	76.17	76.70

*n* = 3 per concentration, µg/kg refers to the values of feces; µg/L refers to the values of plasma and urine.

**Table 3 toxins-16-00051-t003:** Plasma toxicokinetic parameters following single ZEN oral in donkeys.

Toxicokinetic Parameters	Value
Body weight (kg)	154.88 ± 4.76
ZEN (µg·kg·BW^−1^)	2000
Tmax (h)	0.48 ± 0.10
Cmax (µg·L^−1^)	15.34 ± 5.12
T_1/2_Elim (h)	1.63 ± 0.46
AUC (µg·L^−1^·h)	22.30 ± 2.42
Cl (L·kg·BW^−1^·h^−1^)	95.20 ± 8.01
Vd (L·kg·BW^−1^)	216.17 ± 58.71

Tmax = time at maxima concentration of ZEN in plasma, Cmax = maximum plasma concentration, T_1/2_Elim = terminal elimination half-life, AUC = area under the plasma concentration-time curve, Cl = total plasma clearance, Vd = volume of distribution.

**Table 4 toxins-16-00051-t004:** Amount of zearalenone and metabolites in both feces and urine following a single oral administration of zearalenone.

Parameters	Mass Value (mg)	The Amount of Substance (µmol)
ZEN intake	309.75 ± 9.53	972.94 ± 29.92
ZEN excretion via feces	1.61 ± 0.23	5.06 ± 0.73
α-ZOL excretion via feces	2.96 ± 0.48	9.25 ± 1.49
β-ZOL excretion via feces	3.24 ± 0.82	10.10 ± 2.56
Total ZEN excretion through feces (%)	2.50 ± 0.44	2.49 ± 0.43
Absorption rate (%)	97.50 ± 0.44	97.51 ± 0.43
ZEN excretion via urine	1.31 ± 0.30	4.13 ± 0.94
α-ZOL excretion via urine	2.11 ± 0.59	6.58 ± 1.85
β-ZOL excretion via urine	3.06 ± 0.52	9.56 ± 1.63
Total ZEN excretion through urine (%)	2.11 ± 0.46	2.10 ± 0.46

ZEN excretion through feces (%) = total ZEN excretion through feces/ZEN intake × 100. Total ZEN excretion through feces = sum of ZEN+ α-ZOL + β-ZOL excretion via feces. ZEN excretion through urine (%) = total ZEN excretion through urine/ZEN intake × 100. Total ZEN excretion through urine = sum of ZEN + α-ZOL + β-ZOL excretion via urine. Absorption rate (%) = (ZEN intake − total ZEN excretion through feces)/ZEN intake × 100.

## Data Availability

The data presented in this study are available on request from the corresponding author.

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
