# Peer review of "Toxicokinetics of Zearalenone following Oral Administration in Female Dezhou Donkeys"

_toxins, 2024, doi:10.3390/toxins16010051_

Round 1

Reviewer 1 Report

Comments and Suggestions for Authors

The manuscript entitled "Toxicokinetics of Zearalenone following oral administration in female Dezhou Donkeys" analyze the toxicokinetics of ZEN on Dezhou female donkeys following a single oral dosage  by using liquid chromatography-mass spectrometry. 

It is an article  well-organized and a pleasant read. However, there are a few aspects that could benefit from refinement.

In the UPLC-MS/MS analysis section: is the flow isocratic or gradient flow?

In the 2.1. Method Validation section you can add a chromatogram 

Reviewer 2 Report

Comments and Suggestions for Authors

The manuscript is well-presented and concise. The obtained results can contribute to a better understanding of the toxicokinetics of Zearalenone in Dezhouu donkeys after drug oral administration. Furthermore, the conclusions are in line with the presented results. I would call attention to line 113 on page 5, where Figure 1 was called instead of Figure 2.

Author Response

Thank you very much for your comment and suggestion. We have made changes in the revised manuscript.

Reviewer 3 Report

Comments and Suggestions for Authors

This manuscript describes the toxicokinetics of a single oral dose of zearalenone in female donkeys. Why donkeys were chosen is a bit odd since pigs are more sensitive to zearalenone, but it appears to be a continuation of previous studies by the same group investigating the toxicokinetics of other mycotoxins (DON and OTA) on donkeys, using the same protocols (references 31 and 32). The previous studies used male donkeys while this study used female donkeys, likely due to the effect zearalenone has in causing reproductive disorders. The levels of zearalenone and its derivatives as a function of time are quantitated by UHPLC-MS/MS.

The manuscript is relatively short (13 pages, including 38 references) and the results are summarized in four figures and three tables with Figure 1 being the calibration curves. The introduction is brief (one page) but covers most of what appears in the literature regarding the toxicity of zearalenone. The time course is detailed and the analytes are measured in several body fluids (urine, feces and plasma) but the number of animals used, at 4, is the minimum to obtain good results. This is likely due to the cost of such animals for testing. However, the authors state that the study was approved by their animal welfare ethical committee. The methods and quantitation appear to be adequately validated.  

The discussion adequately explains the results and compares to studies using other animals. The main conclusion is that zearalenone absorption, elimination and excretion parameters are different from other animals, similar to what was reported in the group’s previous studies (references 31 and 32).  The toxin has a wider tissue distribution and prolonged tissue persistence but the authors don’t speculate on how this affects toxicity in this particular animal.

However, there are numerous misspellings, typos and grammatical errors. On many occasions, ZEN is referred to as “ZNE” or “ZEA” (see lines 33, 56, 90, 138), and the metabolites of ZEN, namely ZOL, ZAL and ZAN should be defined either in the abstract or when first mentioned, rather than being defined on page 5, lines 116-117. The authors should proof the manuscript carefully before it can be accepted.  

Specific comments:

Abstract, line 7: grammar – should be “toxic effects” or “toxicity effects” “in”

Line 17: the meaning of this sentence is unclear – please clarify.

Page 1, line 29: what is a “detection ratio”?; line 30 – grammar - ZEN is not a “contaminated mycotoxin” – do the authors mean one of the highest occurring mycotoxins”?; line 33 – “years” should be plural.

Page 2, line 72: “there is little research…”. There is likely no research on ZEN in donkeys before this study.

Page 5, lines 116-117: leave out “While” – otherwise the sentence makes no sense.

Page 6, line 137-138:  should read “The metabolites of ZEN, namely α-ZOL and β-ZOL were detected in the urine”.

Page 7, line 149: “humans” is plural.

Page 8, line 197: “the Vd of pigs ranged between 7.27 and 99”? Clarify this sentence. Line 201 – “tissue” is singular. Line 207 – “labelled”.

Page 9, lines 227-228: “sample” is singular and omit “while”

Comments on the Quality of English Language

There are numerous misspellings, typos and grammatical errors. The authors need to proof the manuscript carefully before it can be accepted.
